# Immobilized covalent triazine frameworks films as effective photocatalysts for hydrogen evolution reaction

Xunliang Hu[1], Zhen Zhan[1], Jianqiao Zhang[2], Irshad Hussain [3] & Bien Tan [1✉]

Covalent triazine frameworks have recently been demonstrated as promising materials for photocatalytic water splitting and are usually used in the form of suspended powder. From a practical point of view, immobilized CTFs materials are more suitable for large-scale water splitting, owing to their convenient separation and recycling potential. However, existing synthetic approaches mainly result in insoluble and unprocessable powders, which make their future device application a formidable challenge. Herein, we report an aliphatic amine-assisted interfacial polymerization method to obtain free-standing, semicrystalline CTFs film with excellent photoelectric performance. The lateral size of the film was up to 250 cm$^2$, and average thickness can be tuned from 30 to 500 nm. The semicrystalline structure was confirmed by high-resolution transmission electron microscope, powder X-ray diffraction, grazing-incidence wide-angle X-ray scattering, and small-angle X-ray scattering analysis. Intrigued by the good light absorption, crystalline structure, and large lateral size of the film, the film immobilized on a glass support exhibited good photocatalytic hydrogen evolution performance (5.4 mmol h$^{-1}$ m$^{-2}$) with the presence of co-catalysts i.e., Pt nanoparticles and was easy to recycle.

[1] Key Laboratory of Material Chemistry for Energy Conversion and Storage Ministry of Education, Hubei Key Laboratory of Material Chemistry and Service Failure, School of Chemistry and Chemical Engineering, Huazhong University of Science and Technology, Luoyu Road No. 1037, 430074 Wuhan, China. [2] National Facility for Protein Science in Shanghai, Zhangjiang Lab, Shanghai Advanced Research Institute, CAS, No.333, Haike Road, Shanghai, Shanghai 201210, China. [3] Department of Chemistry and Chemical Engineering, SBA School of Science and Engineering (SSE) Lahore University of Management Sciences (LUMS), Lahore Cantt 54792, Pakistan. ✉email: bien.tan@mail.hust.edu.cn

Covalent triazine frameworks (CTFs) with aromatic triazine linkages are a subclass of covalent organic frameworks (COFs). CTFs are constructed by covalently linking light elements (C, N, and H), which exhibit intriguing physical/chemical properties such as high porosity, high nitrogen content, and good thermal/chemical stability[1−3]. These unique properties endow CTFs with great prospects in various applications including gas separation and storage, energy storage, and photo/electro/thermo-catalysis[4−9]. In the development of CTFs, substantial improvement has been made in the preparation method and crystallinity, but the processability problem still limits its practical applications.

It is a great challenge to obtain CTFs under mild conditions because of high energy barrier of aromatic nitriles trimerization reaction. The high temperature of ionothermal method causes partial carbonization of the CTFs structure and the materials are obtained in the form of black powders[10]. CTFs prepared by acid catalysis need strong and corrosive acid such as trifluoromethylsufonic acid, which is not suitable for acid-sensitive building blocks and practical application[11]. In 2017, our group reported a pioneering low-temperature polycondensation method for CTFs fabrication under ambient conditions; however, the CTFs obtained were amorphous[12]. Recently, our group developed strategies such as in situ oxidation and controlling monomer feeding rate to prepare crystalline CTFs[7,13].

Crystalline CTFs possess extended conjugated structure in two dimensions and are attractive for photocatalytic hydrogen evolution. Till now, most of the reported water splitting systems are based on suspended CTFs that are often used in small batches and hard to recycle. To address this issue, film-based immobilized photocatalytic systems have been developed such as panel reactor and photo-electrochemical reactors[14,15]. Compared with suspended powder, the film-type photocatalysts have inherent advantages as follows: (1) better photocatalytic performance ascribing to less light scattering and enhanced light absorption to generate more charge carriers; and (2) easy to recycle and scalable, as they can be easily integrated into immobilized photocatalytic system[16−19]. Therefore, the preparation of CTFs films is important for their practical photocatalytic application. However, CTFs prepared by strategies as mentioned above are insoluble powders and hard to process and load on supports[7,10−13]. There are only a few reports for the preparation of CTF-based films and all of them were prepared by the superacid catalytic method[4,6,20−23]. The thickness, lateral size, and crystallinity of such films, however, are difficult to control. Hence, it is still a challenge to fabricate CTFs film with large lateral size and controllable thickness under mild reaction conditions.

Interfacial polymerization, which usually occurs at liquid/liquid interface or liquid/air interface is an effective method to prepare continuous COF membranes or films[24−38]. As analog of COFs, CTFs film may also be obtained through interfacial polymerization. An important prerequisite for interfacial polymerization is that the monomer/monomer or monomer/catalyst should be soluble in two different phases. Typically, the two phases are water/dichloromethane for liquid/liquid system and water/air for liquid/air system. Water is a necessary component in both of these systems. However, the poor solubility of most of COF or CTF monomers in water hampers their interfacial polymerization.

Ogata et al.[39] have reported that ultrathin films of linear aromatic polymers can be prepared at air/water interface using imine as a precursor. It is speculated that we can transform a monomer into a metastable precursor through dynamic reaction and created a stable interface by suspending that precursor in an organic solvent. The precursor released the monomer at the interface to participate in reaction and, consequently, the reaction was confined at interface. This approach offers a promising strategy to avoid the requirement of water phase in interfacial polymerization systems.

In this work, we report a practical and efficient approach for the preparation of free-standing, semicrystalline CTFs film with large lateral size and controllable thickness by aliphatic amine-assisted interface polymerization. In this case, the dimethylsulfoxide (DMSO)-soluble aldehyde monomer was transformed into DMSO-insoluble imine precursor by reacting with n-hexylamine. The imine precursor can spread at the surface of DMSO guiding the initial arrangement of aldehyde monomers to generate

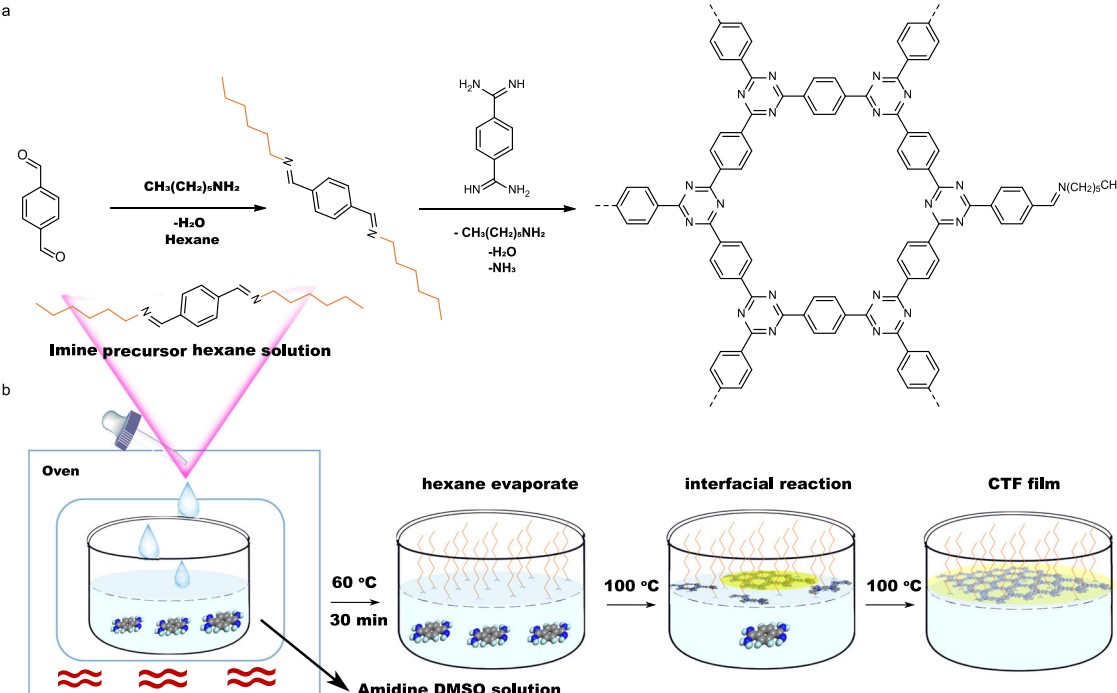

**Fig. 1 Scheme of CTF film synthesis. a** Reaction steps. **b** Synthetic procedure for the fabrication of CTF film on DMSO surface assisted by imine precursor.

DMSO/air interface at the same time. In this way, the polymerization reaction was confined at the interface instead of a homogeneous reaction[7,12]. The semicrystalline structure was probed by high-resolution transmission electron microscope (HR-TEM), powder X-ray diffraction (PXRD), grazing-incidence wide-angle X-ray scattering (GIWAXS), and small-angle X-ray scattering (SAXS) analysis. The film with the large lateral size can be easily loaded on supports as immobilized photocatalyst. In this context, film loaded on glass was employed to test photocatalytic hydrogen evolution reaction (HER) performance. The immobilized photocatalysts with Pt nanoparticles (Pt NPs) exhibit 5.4 mmol $h^{-1}$ $m^{-2}$ (10.2 mmol $h^{-1}$ $g^{-1}$) rate for HER under visible light. This work highlights controlled synthesis of freestanding, semicrystalline, and large-size CTF film based on organic solvent/air interfacial polymerization and provides a rational approach to prepare immobilized photocatalyst, promoting the development of CTFs and COFs films for such application.

## Results and discussions

**Preparation of CTF film.** Typical synthesis of CTFs powders involves a homogeneous condensation reaction between amidine and aldehyde in DMSO[12]. To synthesize CTF film, first, imine precursor was prepared by reacting aldehyde and *n*-hexylamine in hexane (Fig. 1a). Because of the weak polarity of the long carbon chain, imine precursor float at the surface of DMSO layer to generate an interface (Supplementary Fig. 1). The structure of precursor was confirmed by $^1$H-NMR (Supplementary Fig. 2) and Fourier transform infrared (FT-IR) spectroscopy (Supplementary Fig. 3). Peak around 7.8 p.p.m. can be assigned to C-H of benzene ring, peak around 8.3 p.p.m. to C-H of imine bond, and the peaks from 3.7 to 0.9 p.p.m. can be assigned to C-H of aliphatic chain (Supplementary Fig. 2). In FT-IR spectrum, peak of the amino group (3300 $cm^{-1}$) in hexylamine was disappeared and the appearance of a new peak at 1645 $cm^{-1}$ confirmed the formation of imine bond (Supplementary Fig. 3). Amidine monomer and $Cs_2CO_3$ were dissolved in DMSO and then imine precursor was added dropwise on the top of the DMSO layer, while keeping the temperature at 60 °C for 30 min to evaporate hexane. The unstable imine precursor under these conditions was hydrolyzed to the aldehyde that participated in the polymerization reaction (Fig. 1b). The reaction was kept at 100 °C in an oven for 72 h, resulting in the formation of CTF film.

**Characterization of CTF film.** The as-prepared film was freestanding, transparent, soft, and flexible (Fig. 2a–c). The lateral size was up to several hundred square centimeters, which can be adjusted by the reactor size (Supplementary Fig. 4 and Video 1). When transferred to a 4-inch $SiO_2$/Si wafer, it covered the substrate without any noticeable defects (Fig. 2a). The film was dispersed in ethanol for sample preparation for HR-TEM analysis (Fig. 2d–f and Supplementary Fig. 2). HR-TEM images showed that the film comprised nanosheets with lengths ranging from several hundred nanometers to tens of micrometers. The clear lattice fringe with an interplanar lattice space of 0.42 nm corresponds to the (300) atomic plane (Fig. 2e)[13]. The *a* value of the unit cell was calculated to be 14.55 Å based on HR-TEM image.

Selected area electron diffraction (SAED) analysis (Fig. 2f) further confirmed the structure of the film. SAED performed on a single-crystalline domain display hexagonal diffraction pattern with nearest reflections at 2.32 $nm^{-1}$ correspond to (300) interlayer plane, reflections at 4.00 $nm^{-1}$ correspond to (6$\bar{3}$0) interlayer plane (Fig. 2f and Supplementary Fig. 6). The unit cell was assigned to $a = 14.9$ Å, $\gamma = 120°$ (Supplementary Table 1). It must be pointed out that some parts of the film were amorphous

and it was difficult to figure out the exact ratio of the crystalline and amorphous areas, so it is indeed a semicrystalline film. For atomic force microscopic (AFM) characterization, the CTF film was dispersed in ethanol and dropped on a mica plate. AFM images indicated that the film comprised large nanosheets with sizes beyond 10 μm and the thicknesses of ca. 4 nm (Fig. 2g and Supplementary Fig. 7). In general, the reported lateral size of nanosheets in CTF powders prepared by the homogeneous reaction is only several hundred nanometers to several micrometers[7,12,13,40]. Notably, such large nanosheets contributed to the formation of film rather than powder.

To elucidate molecular packing of CTF films on a macroscopic scale, PXRD, SAXS/WAXS, and GIWAXS measurement were performed. The diffraction peak in PXRD at around 7.5° was assigned to (100) reflections and a broad peak at 25° was ascribed to the (001) reflections (Supplementary Fig. 8). To get more information of the ordered structure of CTF film, SAXS analysis was performed on a lab X-ray source in transmission mode. The SAXS profile displayed scattering signal at $q = 0.50$ $Å^{-1}$ corresponding to *d*-spacing of 1.256 nm (100) (Fig. 2h, i). However, scattering signal in SAXS was hardly visible because of low intensity of lab X-ray source. Synchrotron SAXS/WAXS showed clear scattering ring near 0.5 $Å^{-1}$ (Supplementary Fig. 9a, c), which can be assigned to the 100 interlayer plane, and scattering ring near 1.7 $Å^{-1}$ in WAXS can be assigned to (001) interlayer plane (Supplementary Fig. 9b, c). Furthermore, GIWAXS measurements were also performed. In Fig. 3a, reflection ring of CTF film at $Q_{xy} = 0.51$ $Å^{-1}$ was observed, which corresponds to (100) interlayer planes. GIWAXS in larger *q*-range is shown in Supplementary Fig. 10. The integrated curve (Fig. 3b) from the Supplementary Fig. 10 showed peaks at around 0.9 and 1.0 $Å^{-1}$ corresponding to (110) and (200) planes, respectively. The resulting unit cell was assigned to $a = 14$ Å, $\gamma = 120°$, and interlayer spacing was 3.5 Å, agreeing well with SAED results and theoretical structure (Supplementary Tables 1 and 2).

Chemical composition of the film was characterized by FT-IR (Fig. 4a), solid-state NMR spectroscopy (Fig. 4b), Raman (Fig. 4c), and X-ray photoelectron spectroscopy (XPS) (Fig. 4d, e). The presence of peaks at 1517 $cm^{-1}$ (C = N stretching vibration) and 1353 $cm^{-1}$ (C-N stretching vibration) in FT-IR spectra confirmed successful formation of the triazine rings (Fig. 4a)[7,12,13,40]. Peaks near 2900 $cm^{-1}$ could be assigned to terminal *n*-hexylamine groups[39]. $^{13}$C-NMR spectra confirmed the presence of $sp^2$ carbons of triazine ring (170 p.p.m.) and benzene rings (138 and 128 p.p.m.) (Fig. 4b)[12,13,40]. Small peaks at around 10–30 and 116 p.p.m. can be assigned to the terminal aliphatic amine and C = O of aldehyde groups, respectively. Raman spectra showed the disappearance of characteristic aldehyde C = O stretch peak at 1693 $cm^{-1}$ and the emergence of new peaks at 1426 and 1514 $cm^{-1}$, demonstrating the successful transformation of aldehyde monomers into triazine polymers in CTF films (Fig. 4c).

Two peaks in XPS spectra are assigned to carbon of N=C−N (287.3 eV) and C=C (285.0 eV) (Fig. 4d). Peaks at around 398.9 eV can be assigned to nitrogen of triazine ring. Peaks at 399.8–400.0 eV correspond to the pyrrolic nitrogen resulting from the partial decomposition of CTFs (Fig. 4e) and a peak at 403 eV corresponds to the pyridine *N*-oxide nitrogen, which had also been observed and reported in previous work[12,13].

Furthermore, elemental analysis showed that the ratio of carbon and nitrogen contents was close to the theoretical values (Supplementary Table 3). The Brunauer-Emmett-Teller (BET) surface area calculated from the nitrogen adsorption and desorption isotherms at 77.3 K of the films (Fig. 4f) was found to be 110 $m^2$ $g^{-1}$ and the pore size was around 1.1 nm, confirming the formation of CTF[12].

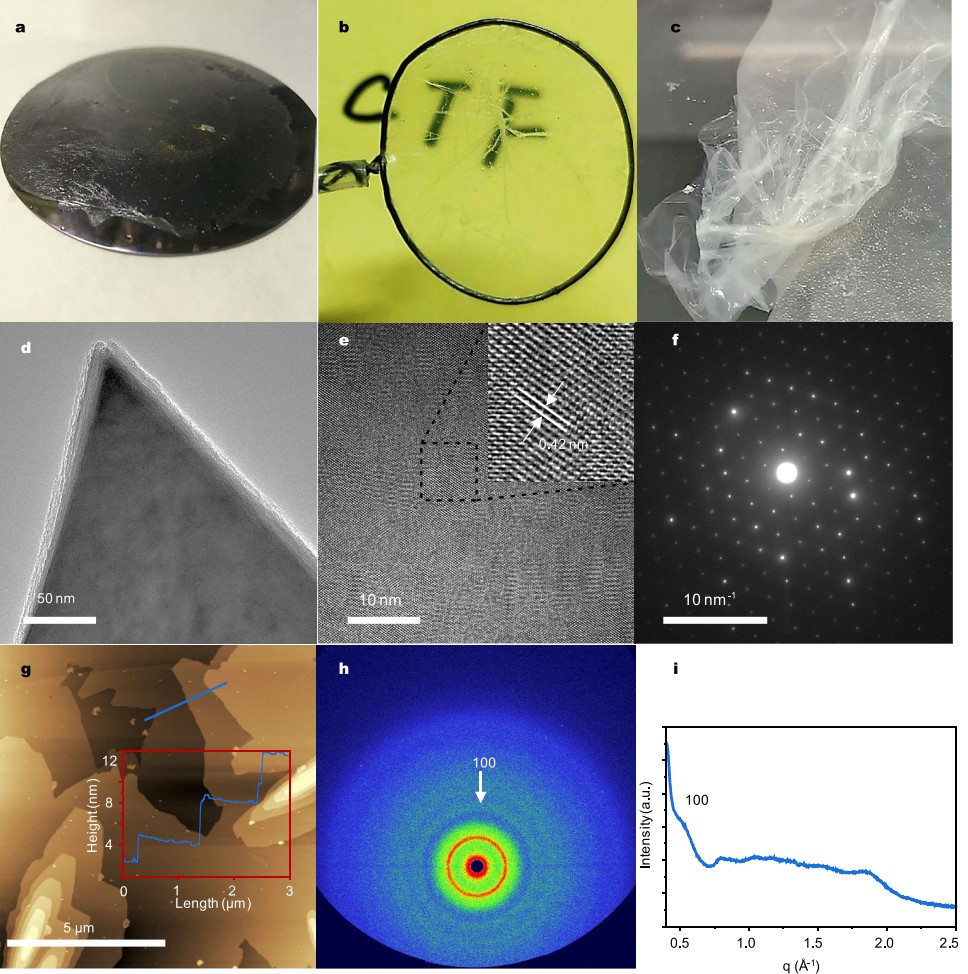

**Fig. 2 Morphology characterization. a** CTF films on 4-inch 300 nm $SiO_2$/Si wafer. **b** The free-standing and transparent CTF film. **c** CTF film immersed in the solvent. **d**, **e** HR-TEM images. **f** SAED image. **g** AFM image. **h**, **i** 2D SAXS image and their corresponding 1D SAXS profiles of the CTF film.

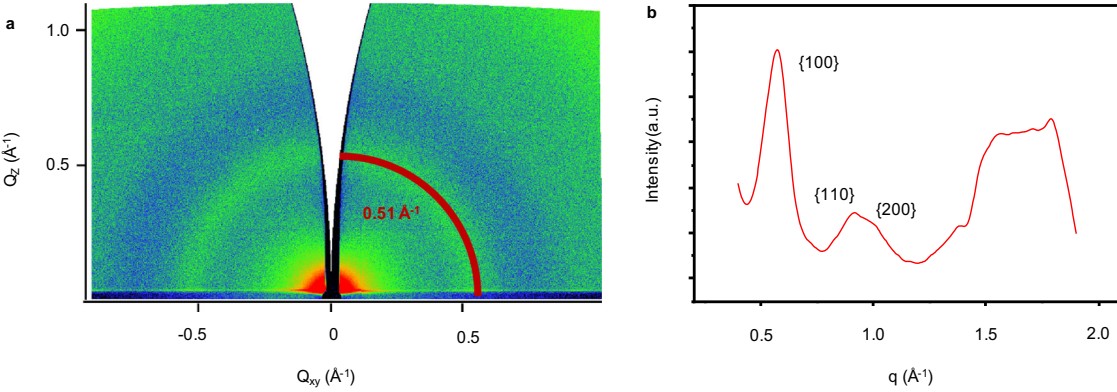

**Fig. 3 GIWAXS analysis of the CTF film. a** GIWAXS image of CTF film on 300 nm $SiO_2$/Si ($q$-range from 0 to ~1.0 Å$^{-1}$). **b** In-plane projection of GIWAXS data from Supplementary Fig. 9 ($q$ ranging from 0 to ~2.0 Å$^{-1}$).

**CTF film thickness regulation**. Controllable film thickness is of great importance for molecular separation and photochemical applications[41]. In this context, we obtained CTF films with thickness ranging from about 150 to 500 nm by adjusting the amount of imine precursor (from 0.017 to 0.050 mmol) and concentration of amidine (from 2.1 to 6.3 mM). The film morphology and thickness was determined by scanning electron microscope (SEM) and AFM. SEM images (Fig. 5a, amidine: 6.3 mM) showed no significant cracks or big particles and the

surface was smooth (Fig. 5b), which confirmed the stability of interface. The film thickness observed in SEM was about 491 nm (Fig. 5c).

Elemental mapping revealed that both carbon (red) and nitrogen (green) atoms were distributed homogeneously (Fig. 5d–f). Decreasing concentration of amidine to 4.2 mM (Supplementary Fig. 11) or 2.1 mM (Supplementary Fig. 12), the films could still cover copper grids without significant cracks (Supplementary Figs. 11a, b and 12a). However, some particles

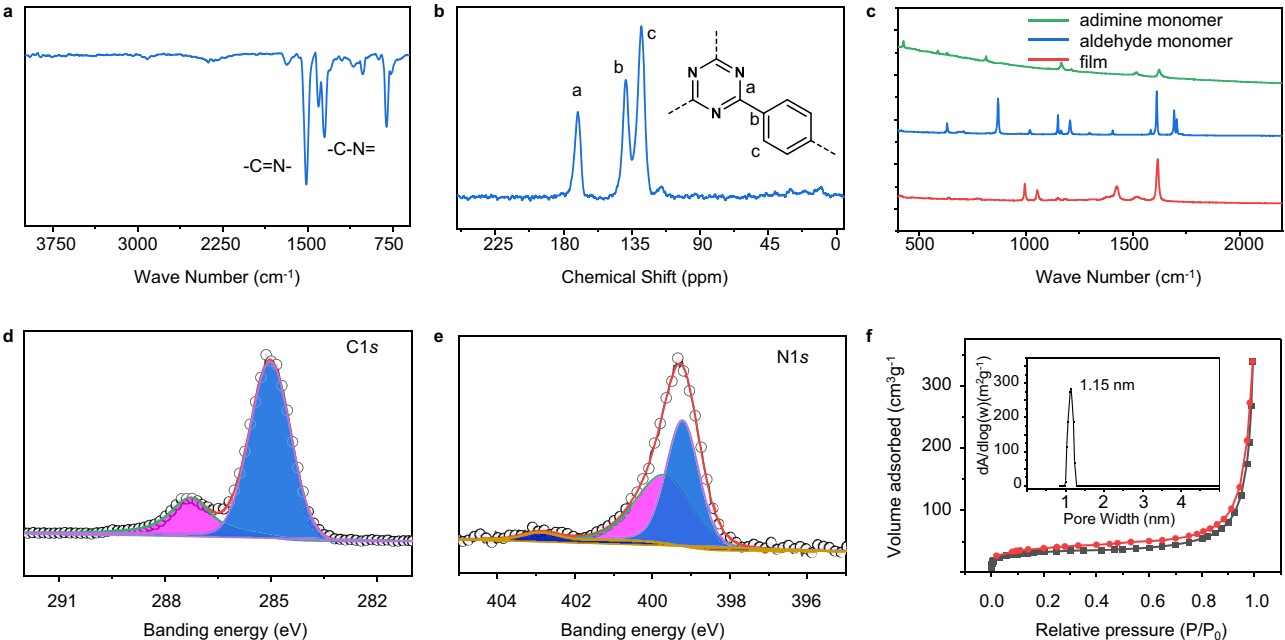

**Fig. 4 The structure characterization of the CTF film. a** FT-IR spectra of the CTF films. **b** $^{13}$C-NMR spectra of the CTF films. **c** Raman spectra of monomers and CTF films. **d**, **e** X-ray photoelectron spectrum of the CTF films. **f** N$_2$ adsorption and desorption isotherms (77 K); insert: the pore size distributions.

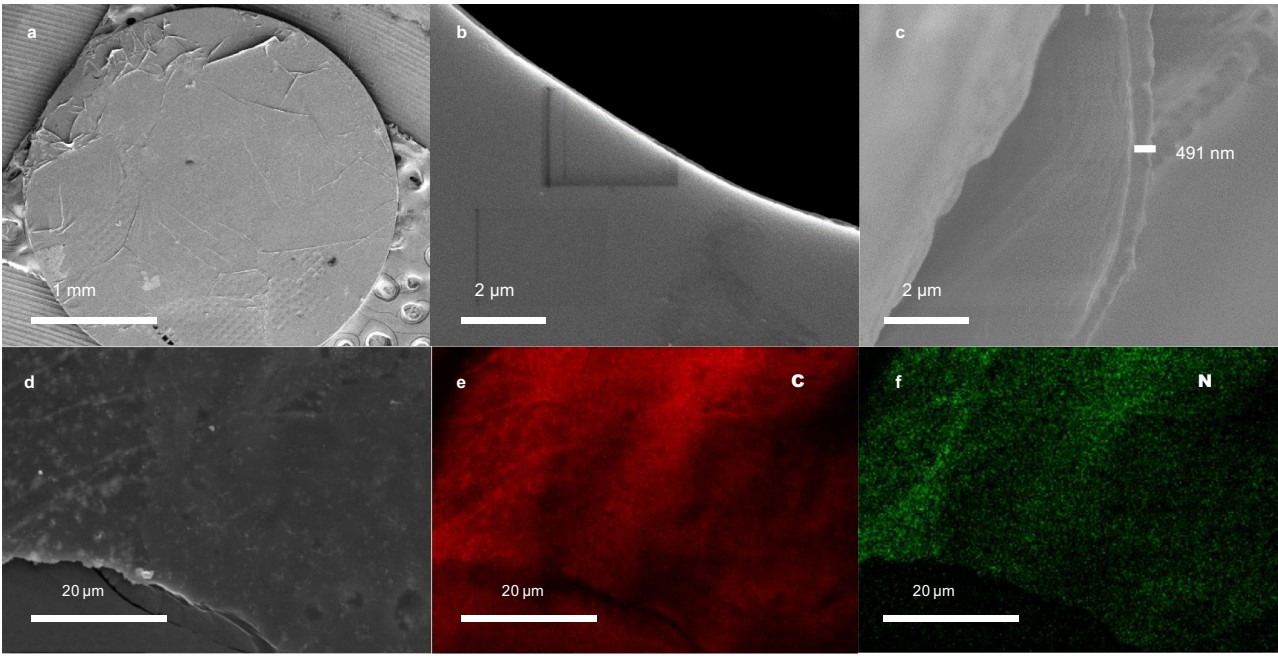

**Fig. 5 The morphology analysis of CTF film. a**, **b** SEM images of the film supported on TEM copper grid. **c** Cross-section SEM image. **d** SEM image and element mapping: **e** carbon; **f** nitrogen. The CTF film was prepared using 6.3 mM amidine.

can be observed at the edge or surface (Supplementary Fig. 11c and 12b, c). It appeared that the CTFs particles were grown on the film, which could not be rinsed off in the workup process. This phenomenon was also observed in previous work, which may be caused by interfacial fluctuations[29,41]. AFM images indicated that the thickness of the films range from 540 to 30 nm (Supplementary Fig. 13–16), which is in agreement with cross-sectional SEM images. Defects could be observed from film with much lower thickness (Supplementary Fig. 16), which might be caused by irreversible polymerization reaction and low monomer concentration. These defects might impact their application in molecular separation. Therefore, it can be concluded that the

concentration of monomers played a vital role in formation of CTFs film and their thickness.

**Photocatalytic HER performance study.** Optical and electronic properties of CTFs film were investigated by ultraviolet-visible (UV-vis) absorption spectroscopy and electrochemical characterization. Solid-state UV-vis diffused reflectance spectrum showed strong absorption by CTFs film at wavelength below 500 nm (Fig. 6a). The light absorption range was similar to that of crystalline CTF powder prepared via benzylamine monomer (160 °C)[40], but narrower than that of crystalline CTF powder

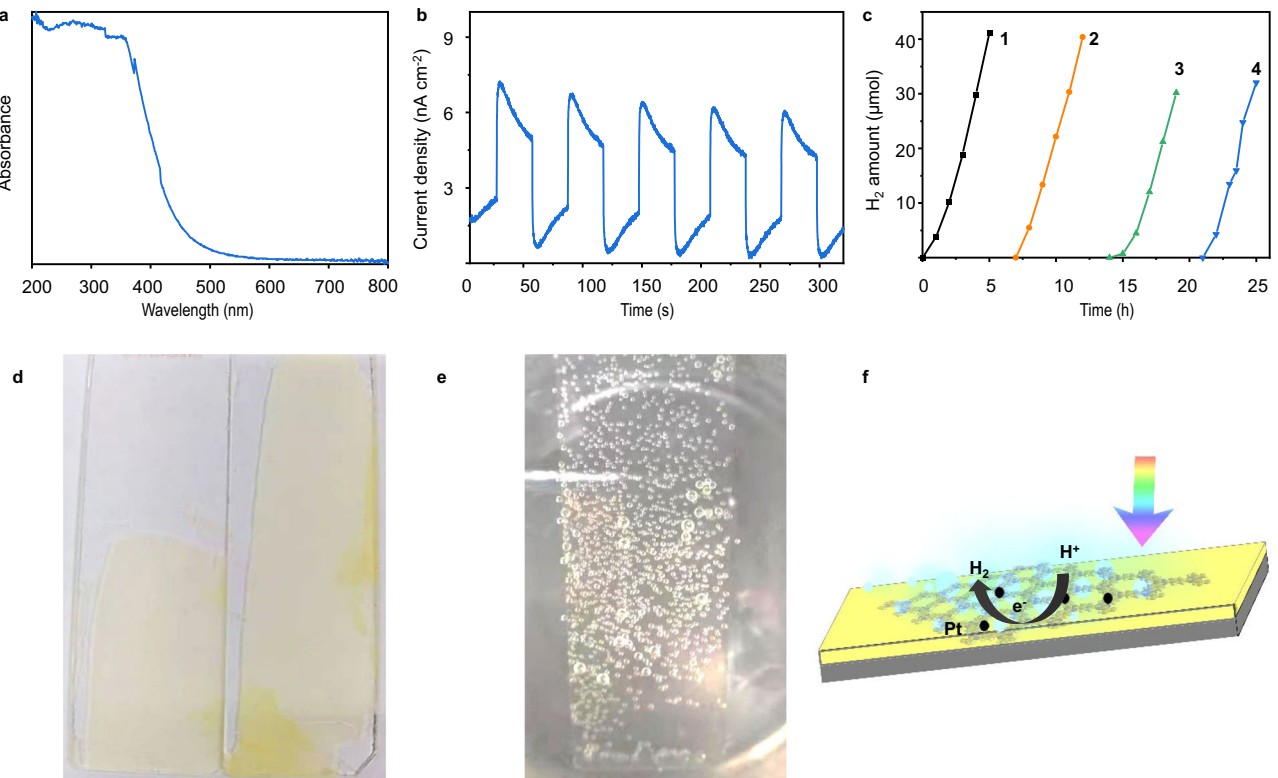

**Fig. 6 Photoelectric and photocatalytic HER performance. a** UV-visible light absorption spectrum of CTF film. **b** The photocurrent curve. **c** Photocatalytic HER performance of film on glass in four cycles under visible light (>420 nm). **d** Images of CTF film loaded on glass. **e** Image of CTF film loaded on glass for photocatalytic HER. **f** Schematic diagram of the photocatalytic process.

prepared at higher temperature (180 °C)[7]. The high reaction temperature may cause higher polymerization and conjugation degree leading to a broader light absorption range. The film on Indium Tin oxide (ITO) glass exhibited current response behavior under visible light irradiation (>420 nm), indicating charge transfer and separation ability (Fig. 6b). According to Mott–Schottky plots, conduction band minima (CBM) of CTF film was estimated to be −0.80 V (Supplementary Fig. 17). The CBM was more negative than the potential of $H^+/H_2$ (−0.41 V NHE at pH 7.0), which meant the energy level was suitable for the photocatalytic proton reduction reaction.

As an attempt, a simple device for photocatalytic HER performance test was fabricated by transferring the film (~500 nm thick) onto a glass slide (size about 19 $cm^2$) (Fig. 6d, e). The photocatalytic hydrogen evolution activity of CTF film was investigated under visible light (>420 nm) with Pt NPs as cocatalyst (Pt source is $H_2PtCl_6 \cdot 6H_2O$), and triethanolamine (TEOA) as a sacrificial agent. After irradiating for 1 h, significant $H_2$ gas bubbles were clearly observed on surface of glass (Fig. 6e). HER rate was calculated to be about 5.4 mmol $h^{-1}$ $m^{-2}$ (equal to 10.2 mmol $h^{-1}$ $g^{-1}$) (Fig. 6d). These results proved that the film could be integrated into photochemical devices with the retention of intrinsic properties. Immobilized photocatalysts are endowed with special advantages, i.e., easy recycling and good performance, which are critical for practical application. After photocatalysis, the glass slides can simply be taken out of the reactor instead of the energy and time-consuming separation process required in suspended powder system[16]. Long-term hydrogen evolution experiments showed steady hydrogen production over four cycles, indicating that the film was stable under the reaction conditions ($\lambda > 420$ nm, 10% TEOA) (Fig. 6d) with a little decrease in $H_2$ evolution after two cycles. It might be due to an impurity adsorbed on the film surface or a part of film might have peeled off from glass during the washing. The photochemical

performance of CTFs film was comparable or even better than that of some of the previously reported CTFs and COFs powder (Supplementary Table 4). The performance of this CTF film-based photocatalytic system (0.12 L $h^{-1}$ $m^{-2}$) was further compared with that of the carbon nitride films (0.19 L $h^{-1}$ $m^{-2}$) and FS-COF film (0.36 L $h^{-1}$ $m^{-2}$) (Supplementary Table 5). Overall, the CTFs film was easily integrated into a photochemical device that was found to be very promising for its practical application in photocatalytic HER.

**Long-term photocatalytic experiments and electrochemical properties.** Long-term stability of the photocatalytic performance and the structure of the CTF film after the photocatalytic experiment were also investigated. The film without Pt co-catalyst showed negligible $H_2$ evolution rate, which confirmed the critical role of Pt in the photocatalytic HER process (Fig. 7a). Photocatalytic experiment (100 h; Fig. 7b) showed that the film exhibited stable $H_2$ evolution rate in 60 h, but the rate decreased after 60 h and remained photocatalytic active and stable till 100 h. After adding fresh TEOA solution at 100 h, the film still showed stable $H_2$ evolution for another 50 h. These results show the CTF film is stable in a relatively long photocatalytic process. The decrease of HER rate in the long-term photocatalytic experiments had also been observed previously in the film-based inorganic photocatalytic systems. The possible reason may be the backward reactions or the agglomeration/detachment of co-catalysts. Apparent quantum yield (AQY) was found to be 0.11% at 420 nm and 0.01% at 500 nm (Supplementary Table 6). AQY values of some typical COFs reported so far are also summarized in Supplementary Table 4. However, AQY cannot be fully comparable between different COFs, because it was not measured using the same protocol[42]. Furthermore, we investigated the electrochemical properties of the Pt@film (the film after the

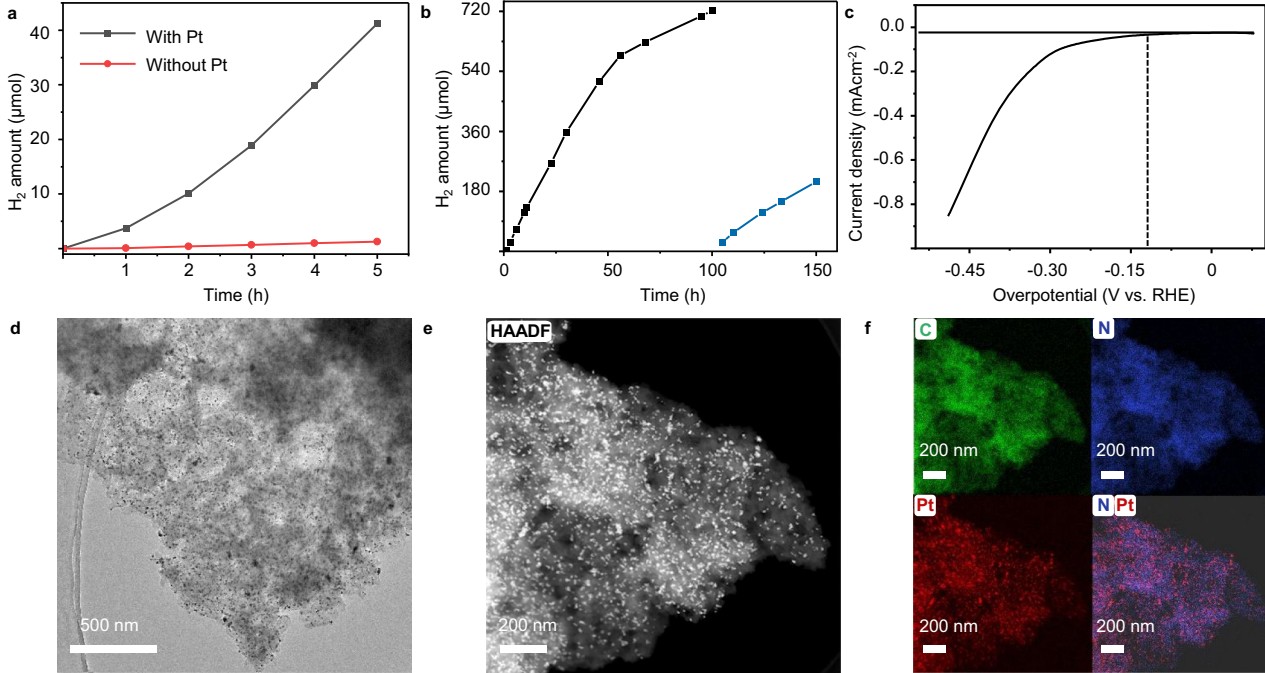

**Fig. 7 Long-term photocatalytic HER performance, electrochemical properties, and HR-TEM analysis of the film after photocatalytic experiments.**
**a** HER performance of the film with and without Pt-co-catalysts. **b** Long-term photocatalytic HER performance (100 h experiment; then fresh TEOA was added at 100 h for another 50 h photocatalytic experiment). **c** HER polarization curves in 0.1 M KOH solution. **d** HR-TEM image of CTF film after photocatalytic experiment. **e** High-angle annular dark-field scanning transmission election microscope (HAADF-STEM) image of Pt NP-loaded CTF films. **f** EDS mapping; green: C; blue: N; red: Pt.

photocatalytic experiment) on ITO support. The overpotential of the Pt@film was 120 mV vs. RHE (Fig. 7c). However, the current density was low because of the low Pt loading (2.0 wt% by Inductive Coupled Plasma Emission Spectrometer - ICP). The Faraday efficiency was 92% when using a constant voltage −0.35 V vs. RHE, which confirmed the good selectivity of $H_2$ evolution (Supplementary Fig. 18 and Supplementary Table 7).

HR-TEM images show abundant Pt nanoparticles distributed uniformly on the film (Fig. 7d–f). The Pt content was found to be 2.0 wt% based on ICP analysis. The carbon and nitrogen distribution from elemental mapping and energy dispersive spectrometer with TEM revealed that both carbon (red) and nitrogen (green) atoms are distributed homogeneously in the film. The FT-IR and Raman spectra of films before and after photocatalytic experiments showed no difference, indicating no change in the chemical nature of the films. The FT-IR, Raman spectra, and long-term stable photocatalytic performance confirmed that the films were stable in the photocatalytic experiments (Supplementary Figs. 19 and 20). In addition, the TEM images obtained from the post-photocatalysis sample (Supplementary Figs. 21–25) demonstrate the retention of crystalline domains in addition to the platinum nanoparticles that are formed in situ during photocatalysis, which confirmed the order structure also keeps stable in the long-term photocatalytic process.

## Discussion

The development of CTFs is currently facing three major challenges as follows: (1) mild and efficient synthesis approach, which impacts chemical structure and band structure of results materials; (2) crystallinity, which limits charge transfer and photocatalytic performance; and (3) processability, which limits their practical applications. In the past 10 years, substantial improvement has been made in the synthesis approach and crystallinity of such

materials, but their processability has always been limiting their practical applications. Herein, we have demonstrated the successful synthesis of free-standing, semicrystalline, large-area CTF film via an aliphatic amine-assisted interfacial polymerization. By tuning the concentration of monomers, the thickness can be easily regulated from 30 to 500 nm. The conjugated network and large lateral size offered an excellent model to investigate their photocatalytic HER performance as immobilized photocatalysts. The HER rate was found to be as high as 5.4 mmol $h^{-1}$ $m^{-2}$ (equal to about 10.2 mmol $h^{-1}$ $g^{-1}$) under visible light in the presence of co-catalyst (Pt nanoparticles). We believe that the proposed air/organic solvent interfacial polymerization strategy offers a rational approach to prepare immobilized photocatalyst and can also address the shortcoming of insufficient solubility of monomers in previously reported interfacial strategies. This strategy also offers a significant development to overcome the processability of CTF films. These findings, therefore, offer a significant development in the formation of CTFs and COFs films for their potential integration in future devices for applications in photoelectric, sensing, separation, and energy technologies.

## Methods

**Synthesis of imine precursor.** Imine precursor was synthesized using a reported method[32]. Typically, 6.7 mg terephthalaldehyde (0.05 mmol) were dissolved in 1.0 mL hexane in a vial, followed by the addition of 13.2 μL n-hexylamine (0.10 mmol) into the vial. The vial was stirred at 60 °C for 2 h.

**Preparation of lower layer solution.** Terephthalamidine dihydrochloride (23.5 mg, 0.10 mmol) and cesium carbonate (33.0 mg, 0.10 mmol) were dissolved in 16 mL DMSO and stirred at 100 °C for 30 min. The obtained mixture was used as a lower layer (6.3 mM).

**Typical procedure for the preparation of the CTFs Film.** The lower layer solution was added to a glass culture dish (9.0 cm in diameter) that was placed in an oven. The imine precursor was added on the top of the lower layer solution that was uniformly spread on the DMSO surface. After keeping it at 60 °C for 30 min to

let the hexane evaporate, the temperature of the oven was increased to 100 °C and kept for 72 h. Then, 200.0 mL water was added into the dish, to let the film float up, and then the film was immersed in DMF, ethanol, and water to remove unreacted monomers and solvent. After that, the film can be transferred to other substrates or taken out by pipette for the characterization or photocatalytic application. The thickness of the film was about 500 nm (as measured by SEM, 6.3 mM amidine).

The 340 nm film: Lower layer: 4.2 mM solution of terephthalamidine dihydrochloride in DMSO (16.0 mL), 33.0 μmol terephthalaldehyde, and 66.0 μmol $n$-hexylamine in 1.0 mL hexane.

The 150 nm film: 2.1 mM solution of terephthalamidine dihydrochloride in DMSO (16.0 mL), 16.0 μmol terephthalaldehyde, and 32.0 μmol $n$-hexylamine in 1.0 mL hexane.

The 30 nm film: 0.21 mM solution of terephthalamidine dihydrochloride in DMSO (16.0 mL), 1.6 μmol terephthalaldehyde, and 3.2 μmol $n$-hexylamine in 1.0 mL hexane.

*Photocatalytic experiments*. The photocatalytic performance was measured under the irradiation of visible light (>420 nm) at 15 A current with 300 W Xe lamp (Beijing Perfect Light Co. Ltd, PLS-SXE300). The diameter of the photoreactor was 7.8 cm. The whole photocatalytic process was kept at room temperature (25 °C) with a light intensity of 130 mW cm$^{-2}$. The hydrogen production was determined by gas chromatography (SHIMADZU, GC-2014C). The CTF film (500 nm thick) was transferred to a glass slide with 19 cm$^2$ area that was dried in an oven at 60 °C for 24 h, then the glass was immersed in 20 mL ethanol that contains 10 μL H$_2$PtCl$_6$•6H$_2$O (10 mg mL$^{-1}$) and kept for 24 h to load the H$_2$PtCl$_6$. The Pt content after photocatalytic HER experiment was found to be 2.0 wt.% based on ICP analysis. The glass was then immersed in 100 mL TEOA aqueous solution (10 vol %, v/v) for photocatalytic HER. After the photocatalytic testing, the film was scraped off and weighed (0.9 mg). Considering the weighing error, the weight was considered as 1.0 mg to calculate the HER rate. The detailed experimental procedure of cyclic experiments were as follows: (1) the film-based catalysts were taken out, washed with deionized water thoroughly, and dried at 60 °C; (2) fresh TEOA solution (10 mL TEOA + 90 mL deionized water) was added without Pt catalyst in the photocatalysis reactor; (3) washed and dried film catalysts were placed in a photoreactor; and (4) the normal photocatalytic experiment was carried out.

**Long-term photocatalytic experiment**. The photocatalytic system was illuminated by light for 100 h, followed by the addition of fresh TEOA solution, purging with N$_2$ and illumination by light for another 50 h.

*Photo-electrochemical measurements*. The EIS and photocurrent measurements were carried out by an electrochemical workstation (CHI760E) equipped with a conventional three-electrode system. A platinum plate electrode and Ag/AgCl electrode were used as the counter electrode and the reference electrode, respectively. Data were measured at least three times.

## Data availability
All data supporting the findings of this study are available within the article, as well as the Supplementary Information file, or available from the corresponding authors on reasonable request. Source data are provided with this paper.

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

## Acknowledgements

We thank the Analysis and Testing Center, Huazhong University of Science and Technology, for assistance in the characterization of materials. This work was financially supported by funding from National Natural Science Foundation of China (Grant Numbers 22161142005 and 21975086), the International S&T Cooperation Program of China (Grant Number 2018YFE0117300), the HUST Graduate Innovation Funding (Grant Number 2020yjsCXCY028), and Science and Technology Department of Hubei Province (Number 2019CFA008). We thank the staffs from the BL19U2 beamline of National Facility for Protein Science in Shanghai (NFPS) at The Shanghai Synchrotron Radiation Facility, for assistance during data collection.

## Author contributions

B.T. conceived the project and designed the experiments. X.H. performed the experiments and analyzed the data. X.H. and B.T. co-wrote the manuscript. I.H. helped in interpreting the results and improving the manuscript. Z.Z. helped in collecting HR-TEM data (Fig. 6d–f and Supplementary Figs. 21–25). J.Z. helped in collecting synchrotron SAXS/WAXS data.

## Competing interests

The authors declare no competing interests.
