## [Peer Review File · Nature Communications]

Reviewers' Comments:

Reviewer #1:

Remarks to the Author:

Review Report for Nature Communications

The manuscript from Hu et.al. reports the synthesis of large-sized and free-standing thin films of covalent triazine framework (CTF) via an aliphatic amine-assisted interfacial polymerization method. Authors have shown that the CTF films with a lateral size up to 250 cm² and the thickness ranging from 30 to 500 nm can be prepared. The large-sized films with good photoelectric performance have been applied as a semiconductor catalyst for photocatalytic hydrogen generation. Although the values reported for photocatalytic hydrogen evolution (5.4 mmol h⁻¹ m⁻²) are good, the crystallinity of the films obtained is very limited. In addition, basic characterization such as Grazing-Incidence Wide-Angle X-ray Scattering (GIWAXS), Raman spectroscopy analyses, etc. are missing.

Manuscript holds certain novelties about the synthesis of large-sized CTF films and their applications for photocatalytic water splitting, however, it is unclear whether the 2D films are crystalline as the authors try to claim or it is just a random mixture of crystalline and amorphous domains. This is the main reason I would like to withhold my suggestion to recommend publications as it stands. Following are suggestions where the authors could potentially improve and questions regarding the characterizations and properties of these materials:

1. The basic claim that authors have made is the crystallinity of the CTF films, which is not much clear from SAED, PXRD and SAXS profiles. Authors have claimed that SAED confirms the good crystallinity of the films, but not mentioned about the unit cell parameters and other calculations.
2. Also, authors have mentioned that the SAXS profile displayed a scattering signal at $q = 0.50 \text{ \AA}^{-1}$, which is hardly visible. Recently, it has been showcased that even free-standing two-dimensional polymers also show convincing radial integration intensity profiles (Angew. Chem. Int. Ed. 2020, 59, 6028–6036).
3. In contrast to their own reports (Angew. Chem. Int. Ed. 2018, 57, 11968–11972; Angew. Chem. Int. Ed. 2020, 59, 6007–6014), the PXRD profiles for CTF films are very broad and inconclusive for proving the crystallinity. The diffraction peaks corresponding to 100 and 001 reflections are too broad.
4. To prove the claim about the CTF film crystallinity, authors should also provide the Grazing-Incidence Wide-Angle X-ray Scattering (GIWAXS) analyses.
5. The scheme of synthesis is not much clear at first look as steps involved in the synthesis are not much clearly visualized. Until the synthesis section, it is not clear where hexane goes and how heating takes place.
6. The elemental mapping for carbon and nitrogen atoms shown in figure 3 give a hint that distribution of carbon atoms is very uniform, however, nitrogen atoms are not present in the proportion. Roughly, the proportion of nitrogen to carbon should be 1:3, this does not seem the case. Authors should provide the elemental analyses.
7. Are these films porous?
8. The crystalline CTFs synthesized using different methods (Angew. Chem. Int. Ed. 2018, 57, 11968–11972; Angew. Chem. Int. Ed. 2020, 59, 6007–6014) absorb more light in the visible region as compared with CTF films reported herein. Authors should explain about it in the manuscript. On the contrary, authors have claimed that 'The improvement in photochemical performance may be ascribed to the enhanced light absorption of CTF films', which is not completely true.
9. The long-term hydrogen evolution experiments are limited to 5 hours in one cycle. What is happening with the performance beyond the said time (around 15-20 hours)? Are these films stable and active under reaction conditions at longer duration?
10. Although authors claim that there is a little decrease in the hydrogen evolution performance after the second cycle, from figure 4c, it seems that the performance is decreased by more than 20-25%. What is the reason for such low activity just after 10 hours?
11. What is apparent quantum yield (AQY) for the films?
12. The characterization of CTF films after photocatalytic hydrogen evolution experiments are missing. Are the films structurally robust and maintain the structure after photocatalysis

experiments?

13. There are many grammatical and typographical errors.

For example:

- 'In early studies, it is technically difficult' is a grammatically not correct statement.
- The figure captions for all the figures have many errors.
- The statement 'Researchers were faced' is a grammatically not correct statement.

Reviewer #2:

Remarks to the Author:

The manuscript presented by B. Tan et al. is an excellent report on how to prepare large crystalline films of a COF (in this report a covalent triazine framework (CTF)). As explained by the authors in the manuscript, most of the CTF used as photocatalysts in the hydrogen evolution reaction (HER) or carbon dioxide reduction reaction (CO₂RR) are solid powder suspensions, but for their use in photoelectrocatalytic devices, films are a much better solution. In this sense, the material prepared in this manuscript is an excellent step forward in this direction. The synthetic organic approach to obtain the CTF films ground-breaking and it will open many new opportunities to develop this field.

In general, the characterization of the material is thoroughly performed, and the results obtained confirm that the materials were obtained without doubts. There are some minor aspects in the characterization that I think that should be improved, but I will describe them at the end of this report. In any case, they do not take away any merit to the manuscript. The experimental details for the preparation of the materials and the characterization seems to be enough detailed to reproduce the preparation of the films.

My main concern is related to the photocatalytic tests in the HER reaction. The experiments are acceptable, but I think these comments should be addressed:

Page 12: Hydrogen evolution reaction (1 hour) and long-term hydrogen evolution experiments. Authors describe several experiments to determine the applicability of the material as a photocatalyst in HER reaction when assisted with Pt as a co-catalyst. In this regard, I am missing a full characterization of the material after the photocatalytic tests and some further (photo)(electro)chemical experiments:

- 1) what is the fate of the "molecular" Pt co-catalyst? HR-TEM measurements after photocatalysis could show if Pt NPs were formed during photocatalysis.
- 2) After some long-term cycles, I would perform the photocatalytic test with a fresh TEOA solution (without co-catalyst): if Pt NPs were formed, the material should still be active.
- 3) HR-TEM images after catalysis would show if the film was degraded during photocatalysis.
- 4) Instead of cycles, I would perform a single long-term photocatalytic experiment (+100 hours) to check the stability of the material. Moreover, it is not clear the experimental procedure used in each cycle: new TEOA and Pt catalyst in each cycle?
- 5) What happens when the reaction is performed without the addition of the Pt co-catalyst in the first cycle (or in a single 1-hour experiment)?
- 6) Is it possible to check the activity of the "film + Pt" material in an ITO electrode without light or TEOA and applying a potential to check the electrochemical properties of the material? I expect it to be very active: overpotential for HER could be obtained. Moreover, using a Clark electrode, faradaic efficiencies could be obtained, and this would show that the hydrogen evolution reaction is the only reaction taking place.
- 7) After photocatalytic experiments, and after thoroughly washing the film, ICP analyses of the film will show if undetectable Pt NPs (ultrasmall) or coordination compounds are retained in the film.
- 8) A more detailed comparison with other similar systems should be added. Please, cite and compare the manuscript results with the results described in: doi.org/10.3390/catal11060754

Minor general points:

- 1) Page 3: Sentence in Lines 32-34. Unfinished sentence? Or just an "and" missing?
- 2) Page 4: Sentence in Lines 75-78. Even if I think that the meaning can be understood, it is difficult to read, and I would rephrase the sentence to make it more clear.
- 3) Page 5. Line 102. "n-Hexane" should be written without capital letters.
- 4) Page 6. Scheme 1. In the chemical reaction depicted in the scheme, I would add the by-

products of the reaction in every step (water?).

5) Page 6. Scheme 1. Line 107. "a) possible reaction mechanism". I would rather describe them as "reaction steps" (or something similar) rather than "reaction mechanism".

6) Page 8. Line 132. Space missing at "be14.55".

7) Page 9. Sentence in Lines 155-157. Again, the sentence seems to be unfinished (or an "and" missing".

8) Page 9. Line 159. XPS description seems unfinished. It seems as if an initial paragraph is missing.

9) Page 9. Lines 163-164. Re-write the text in these lines.

10) Page 12. Line 217. "with Pt as cocatalyst". Even if usually referred just as the cocatalysts, the role of Pt in these photocatalytic reactions is crucial and, at least, the actual Pt source (H₂PtCl₆) should be detailed here.

All in all, I think that this manuscript should be accepted for publication if these comments are addressed (or refuted).

Reviewer #3:

Remarks to the Author:

In this paper, Tan et al. developed a smart synthetic strategy to first create a liquid/air interface, and fabricate successfully free-standing, large area, crystalline Covalent triazine frameworks (CTFs) films by this kind of interface. Furthermore, they explored the application of the film as immobilized photocatalysts and showed that the fabricated CTFs film had an excellent performance of photocatalytic hydrogen evolution. From a practical point of view, immobilized CTFs are more suitable for large-scale water splitting applications, however, existing synthetic approaches mainly result in insoluble and non-processable powders, which makes their future device application still a huge challenge. It is indeed highly interesting in this work that an interface is created without a water phase, which provides a rational approach to prepare immobilized photocatalysts, which is a significant landmark for the development of CTF and COF films and their potential integration in future devices for applications in fields as photoelectric, sensing, separation, and energy technologies. After careful evaluation, I would like to recommend this paper for publication in Nature Communications after the authors address the following important points:

1. Page 6, line 104: "Because of the weak polarity of the long carbon chain, the imine precursor was floating at the DMSO layer instead of dissolving in it". Because this process is very important for the creation of liquid/air interface and the preparation of CTFs films, can the authors give further experiment details to describe this process?

2. Interestingly, the CTF film is crystalline. As far as I know, the CTF powder prepared by the low-temperature poly-condensation method is amorphous (ref. 12). Can the author explain what reason cause this difference?

3. The XPS data (figure 2d) should be clarified further, the peak at 403 eV is not fitted in figure 2d, the authors should give more explanation.

4. Did the authors do some special treatment of the films and glass support in the photocatalytic experiments, such as using some adhesive? The decrease of HER performance may be caused by the peel-off of film from the support.

5. The experimental details of the SAXS and PXRD experiments should be described in the manuscript. Please include details of measurement conditions (i.e., powder or film), sample preparation, etc.

6. Some related literature about COFs films (10.1002/anie.202105190; Angew. Chem. Int. Ed. 2021, 60, 14875-14880; etc) can be added in the revised manuscript.

Given below are our responses (in BLUE colour) to the reviewer' comments. The changes to the manuscript and supplementary information are marked in RED colour.

Reviewer 1:

The manuscript from Hu et.al. reports the synthesis of large-sized and free-standing thin films of covalent triazine framework (CTF) via an aliphatic amine-assisted interfacial polymerization method. Authors have shown that the CTF films with a lateral size up to 250 cm² and the thickness ranging from 30 to 500 nm can be prepared. The large-sized films with good photoelectric performance have been applied as a semiconductor catalyst for photocatalytic hydrogen generation. Although the values reported for photocatalytic hydrogen evolution (5.4 mmol h⁻¹ m⁻²) are good. The crystallinity of the films obtained is very limited.

Response: We thank the reviewer for understanding and appreciating the significance of manuscript.

We agree that the crystallinity of the films obtained is still limited. We think there are two reasons for the limited crystallinity: (1) the crystallinity of CTFs powders are limited: compared with covalent organic frameworks (COFs) that are prepared by dynamic reactions, CTFs are prepared using an irreversible reaction that limits their crystallinity (*Angew. Chem. Int. Ed.* 2018, **57**, 11968-11972. *Angew. Chem. Int. Ed.* 2020, **59**, 6007-6014; *Adv. Mater.* 2019, **31**, 1807865); (2) compared with bulk or thicker crystalline COF or CTF, the thin film exhibited poor crystallinity (*Chem* 2018, **4**, 308-317). So it is still a challenge to get crystalline CTF film. Here, we show that the film prepared by an irreversible reaction also can possess a certain degree of crystallinity.

In addition, basic characterization such as Grazing-Incidence Wide-Angle X-ray Scattering (GIWAXS), Raman spectroscopy analyses, etc. are missing.

Response: This is a good suggestion. We have now provided the GIWAXS (**Figure 2**, **Supplementary Figure 10**), and Raman spectroscopy (**Figure 3c**) analyses of CTF films in the revised manuscript.

Manuscript holds certain novelties about the synthesis of large-sized CTF films and their applications for photocatalytic water splitting, however, it is unclear whether the 2D films are crystalline as the authors trying to claim or it is just random mixture of crystalline and amorphous domains. This is the main reason I would like to withhold my suggestion to

recommend publications as it stands. Following are suggestions where the authors could potentially improve and questions regarding the characterizations and properties of these materials:

Response: After more careful analysis, we agree that the 2D films are consisted of both the crystalline and amorphous domains. We have updated this information in the revised manuscript and believe that calling it a “**semi-crystalline film**”, will be more appropriate.

1. The basic claim that authors have made is the crystallinity of the CTF films, which is not much clear from SAED, PXRD and SAXS profiles.

Response: It is indeed difficult to analysis crystallinity of thin films by normal PXRD and SAXS that are carried out in **transmission mode** due to the overlap of the support signal. For example, even the synchrotron SAXS profile of crystalline COF is not very clear (*J. Am. Chem. Soc.* 2017, **139**, 6736 – 6743). Here, we collected the GIWAXS data, which avoided the impact of support, and we could observe much clear scattered signal and the GIWAXS profile showed a clear peak near 0.51 \AA^{-1} , which gives much convincing evidence of the crystallinity.

Authors have claimed that SAED confirms the good crystallinity of the films, but not mentioned about the unit cell parameters and other calculations.

Response: We are sorry for missing calculation of SAED, which has now been added in revised manuscript.

Supplementary Figure 6. SAED pattern of a single-crystalline domain along [001] direction.

Supplementary Table 1 Lattice parameters obtained by DFT calculation, SAED and GIWAXS measurements.

structure	a	b	c	α	β	γ
CTF1 (DFT) ^[1]	14.5	14.5	3.6	90	90	120
SAED	14.9	14.9	N.A	N.A	N.A	120
GIWAXS	14.0	14.0	3.5	N.A	N.A	N.A

2. Also, authors have mentioned that the SAXS profile displayed a scattering signal at $q = 0.50 \text{ \AA}^{-1}$, which is hardly visible. Recently, it has been showcased that even free-standing two-dimensional polymers also show convincing radial integration intensity profiles (Angew. Chem. Int. Ed. 2020, 59, 6028–6036).

Response: The SAXS signal is not very clear. Because the SAXS experiment shown in figure 2i was carried out in **transmission mode** with the incident X-ray beam pointed directly through the sample. The intensity is limited by the thickness of sample, and the signal of sample is overlapped easily by support. For thin film samples, grazing incidence (GISAXS) can increase the signal intensity and eliminate the impact of substrate (Angew. Chem. Int. Ed. 2020, 59, 6028 – 6036).

Here, we obtained the GIWAXS data carried out on lab X-ray source, which shows clear scattering signal at about $q = 0.51 \text{ \AA}^{-1}$; and also convincing radial integration intensity profiles (**Figure 2; Supplementary Figure 10**).

Furthermore, SAXS/WAXS analysis in transmission mode on a synchrotron radiation source was performed, which shows more clear scattering signal at q around 0.50 \AA^{-1} than the SAXS results that was carried out on lab X-ray source. (**Supplementary Figure 9**)

Supplementary Figure 9. The synchrotron SAXS/WAXS measurement of the film. **a** SAXS. **b** WAXS. **c** Integration intensity profiles of the SAXS/WAXS.

3. In contrast to their own reports (Angew. Chem. Int. Ed. 2018, 57, 11968–11972; Angew. Chem. Int. Ed. 2020, 59, 6007–6014), the PXRD profiles for CTF films are very broad and inconclusive for proving the crystallinity. The diffraction peaks corresponding to 100 and 001 reflections are too broad.

Response: We agree that the PXRD profiles for CTF films are very broad. There are two reasons for the broad peaks. (1) The inappropriate characterization methods: conventional X-ray powder diffraction in the characterization of thin films, the incident x-ray beam interacts only with a very limited sample volume (*Rev. Sci. Instrum.* 2010, **81**, 105105), and it is easily impacted by the signal of support, so it is difficult to get good PXRD profiles. (2) The semi-crystalline films also show broad peaks.

Now we have analyzed the CTF films by GIWAXS that confirms their partial crystallinity.

4. To prove the claim about the CTF film crystallinity, authors should also provide the Grazing-Incidence Wide-Angle X-ray Scattering (GIWAXS) analyses.

Response: This is a good suggestion. GIWAXS is perfectly suited for the investigation of the crystallinity of thin films (*Rev. Sci. Instrum.* 2010, **81**, 105105). We have now provided the Grazing-Incidence Wide-Angle X-ray Scattering (GIWAXS) analyses in the revised manuscript (**Figure 2** and Supplementary **Figure S10**).

In the manuscript, we have added the related discussions as follows:

Fig. 2 *a* GIWAXS image of CTF film on 300 nm SiO₂/Si (q range from 0 ~ 1.0 Å⁻¹). *b* In plane projection of GIWAXS data (q range from 0 ~ 2.0 Å⁻¹).

Furthermore, grazing-incidence wide-angle X-ray scattering (GIWAXS) measurements were also performed. In Fig. 2a, reflection ring of CTF film at $Q_{xy} = 0.51 \text{ \AA}^{-1}$ was observed, which corresponds to (100) interlayer planes. GIWAXS in larger q range is shown in supplementary Fig. 10. The integrated curve (Fig. 2b) from supplementary Fig. 10 showed peaks at around 0.9 Å⁻¹, and 1.0 Å⁻¹ corresponding to (110) and (200) planes, respectively. The resulting unit

cell was assigned to $a = 14 \text{ \AA}$, $\gamma = 120^\circ$, and interlayer spacing was 3.5 \AA , agreeing well with SAED results and theoretical structure (Supplementary Table 1 and Supplementary Table 2).

Supplementary Figure 10. GIWAXS measurement of the film.

5. The scheme of synthesis is not much clear at first look as steps involved in the synthesis are not much clearly visualized. Until the synthesis section, it is not clear where hexane goes and how heating takes place.

Response: We have now revised the scheme to better understand the reaction.

Scheme 1. Scheme of CTF's film synthesis. **a** Reaction steps. **b** Synthetic procedure for the film on DMSO surface assisted by imine precursor.

6. The elemental mapping for carbon and nitrogen atoms shown in figure 3 give a hint that distribution of carbon atoms is very uniform, however, nitrogen atoms are not present in the proportion. Roughly, the proportion of nitrogen to carbon should be 1:3, this does not seem the case. Authors should provide the elemental analyses.

Response: The counts per second (CPS) of our previous elemental mapping data are too low, which made the signal of N very weak. Now, we analyzed the samples again by SEM and EDS

for better elemental mapping (Fig. 4d - 4f). From Fig. 4d - 4f, we can observe the uniform distribution of carbon and nitrogen. Meanwhile, the elemental analyses also show that the ratio of nitrogen and carbon is close to the theoretical value (Supplementary **Table 3**).

Supplementary Table 3 EA results of the film.

Methods	N(%)	C(%)	H(%)
Theoretical	21.9	75.0	3.1
EA (before HER)	17.3	68.6	4.7
XPS (before HER)	9.7	74.7	n.a
EDS (after HER)	13.0	73.0	n.a

Fig. 4 The morphology analysis of CTF film. *a, b* SEM images of the film supported on TEM copper grid. *c* Cross-section SEM image. *d* SEM image; and the elemental mapping images: *e* carbon; *f* nitrogen. The CTF film was prepared using 6.3 mM amidine.

7. Are these films porous?

Response: Yes, these films are porous. The nitrogen adsorption and desorption isotherms at 77.3 K of the films are shown in Fig. 3f, the BET surface area is about 110 $\text{m}^2 \text{g}^{-1}$; pore size distribution that was calculated using DFT methods show that the pore size is around 1.1 nm. These data confirm that the films are porous.

8. The crystalline CTFs synthesized using different methods (Angew. Chem. Int. Ed. 2018, 57, 11968–11972; Angew. Chem. Int. Ed. 2020, 59, 6007–6014) absorb more light in the visible region as compared with CTF films reported herein. Authors should explain about it in the manuscript.

Response: The previous CTF powders were prepared at higher temperatures (160 °C or 180 °C), the higher temperature may cause a much higher degree of polymerization and conjugation degree, resulting in broader light absorption. This explanation has now been added in the revised manuscript.

And the light absorption range is similar to that of crystalline CTF powder prepared via benzylamine monomer (160 °C), and narrower than that of crystalline CTF powder prepared at higher temperature (180 °C). The high reaction temperature may cause higher polymerization degree and conjugation degree leading to broader light absorption range.

On the contrary, authors have claimed that ‘The improvement in photochemical performance may be ascribed to the enhanced light absorption of CTF films’, which is not completely true.

Response: For the same materials, compared with powder type, the film exhibits less

light scattering and enhanced light absorption that generates more carriers (*J. Mater. Chem. A* 2021, **9**, 1353-1371; *Chem. Soc. Rev.* 2013, **42**, 2294-2320). We highlight the difference of powder and film, not the powder with different light absorption range. However, many factors may impact the photochemical performance; and this explanation (enhanced light absorption of CTF films) is not suitable, so we have now removed this kind of explanation in the revised manuscript.

9. The long-term hydrogen evolution experiments are limited to 5 hours in one cycle. What is happening with the performance beyond the said time (around 15-20 hours)? Are these films stable and active under reaction conditions at longer duration?

Response: Following this suggestion, we conducted a single long-term (**100 h + 50h**) photocatalytic experiment. As shown in Fig. 6b, the catalytic performance is very stable in the first 60 h; decreases after 60 h, but still active under reaction conditions till 100 h. And after adding fresh TEOA at 100 h, the film has similar HER rate in the following 50 h; so the film is active under reaction conditions as long as 150 h.

Fig. 6 Long-term photocatalytic HER performance, electrochemical properties, and HRTEM analysis of the film after photocatalytic experiments. **a** HER performance of the film with and without Pt-cocatalysts. **b** Long-term photocatalytic HER performance (100 h experiment; then add fresh TEOA at 100 h for another 50 h photocatalytic experiment). **c** HER polarization curves in 0.1 M KOH solution. **d** HR-TEM image of the film after photocatalytic experiment. **e** HAADF-STEM images. **f** EDS mapping; green: C; blue: N; red: Pt.

10. Although authors claim that there is a little decrease in the hydrogen evolution performance after the second cycle, from figure 4c, it seems that the performance is decreased by more than 20-25%. What is the reason for such low activity just after 10 hours?

Response: From Fig. 4c, it seems that the performance decreased significantly. To investigate the stability of performance in HER reaction, we performed a single long photocatalytic experiment (100 h), the performance is very stable in the first 60 h, followed with decrease after 60 h, but still active and stable till 150 h by replenishing TEOA. So, the most likely reason for the decrease of performance after 2 cycles may be due to the peeling off of film from the glass support during the cyclic process.

Furthermore, decrease of performance in long-term cycles has also been observed in some film-based inorganic semiconductor photocatalytic systems for overall water splitting (the performance is about 65% of the initial activity after 40 h; *Joule*, 2018, 2, 2667 - 2680). The decrease of the activity in long-term cycles is generally attributed to the backward reactions or the agglomeration/detachment of co-catalysts.

11. What is apparent quantum yield (AQY) for the films?

Response: The apparent quantum yield for the films has now been calculated and mentioned in the revised manuscript as shown the table below.

Supplementary Table 6 AQY of CTF film using five band pass filters.

λ	P (mW cm ⁻²)	C (1 h)	AQY
420 nm	31.8	4.1 μ mol	0.11%
435 nm	22.3	1.9 μ mol	0.07%
450 nm	19.1	1.0 μ mol	0.04%
475 nm	15.9	0.5 μ mol	0.02%
500 nm	15.9	0.2 μ mol	0.01%

12. The characterization of CTF films after photocatalytic hydrogen evolution experiments are missing. Are the films structurally robust and maintain the structure after photocatalysis experiments?

Response: The FT-IR, and Raman analysis was conducted to confirm that the films

were structurally robust and maintained the structure after photocatalytic experiments. HRTEM images (Supplementary Figure 21 - 25) after photocatalytic experiments demonstrated the retention of crystalline domains. And the stable HER performance for 150 h also indicate that the CTF film is stable in the photocatalytic process.

Supplementary Figure 19. FT-IR spectra of CTF film before and after photocatalytic experiment.

Supplementary Figure 20. Raman spectra of CTF films before and after photocatalytic experiment.

Supplementary Figure 22. TEM images of CTF film after photocatalysis. Bright field images and FFT images. Zone axis [001].

Supplementary Figure 24. TEM images of CTF film after photocatalysis. Bright field images and FFT images. Zone axis [010].

13. There are many grammatical and typographical errors.

For example:

- ‘In early studies, it is technically difficult’ is a grammatically not correct statement.

- The figure captions for all the figures have many errors.
- The statement 'Researchers were faced' is a grammatically not correct statement.

Response: Thanks for your suggestions; we are sorry for these mistakes, and have tried our best to remove such mistakes in the revised manuscript.

'In early studies, it is technically difficult' is a grammatically not correct statement.

- The figure captions for all the figures have many errors.
- The statement 'Researchers were faced' is a grammatically not correct statement.

Response: All such mistakes/typos have now been removed.

Reviewer #2:

The manuscript presented by B. Tan et al. is an excellent report on how to prepare large crystalline films of a COF (in this report a covalent triazine framework (CTF)). As explained by the authors in the manuscript, most of the CTF used as photocatalysts in the hydrogen evolution reaction (HER) or carbon dioxide reduction reaction (CO₂RR) are solid powder suspensions, but for their use in photoelectrocatalytic devices, films are a much better solution. In this sense, the material prepared in this manuscript is an excellent step forward in this direction. The synthetic organic approach to obtain the CTF films ground-breaking and it will open many new opportunities to develop this field.

In general, the characterization of the material is thoroughly performed, and the results obtained confirm that the materials were obtained without doubts. There are some minor aspects in the characterization that I think that should be improved, but I will describe them at the end of this report. In any case, they do not take away any merit to the manuscript. The experimental details for the preparation of the materials and the characterization seems to be enough detailed to reproduce the preparation of the films.

Response: We thank the reviewer for understanding and appreciating significance of this manuscript. We also appreciate the reviewers' comments and suggestions that helped us to significantly improve the manuscript.

My main concern is related to the photocatalytic tests in the HER reaction. The

experiments are acceptable, but I think these comments should be addressed:

Page 12: Hydrogen evolution reaction (1 hour) and long-term hydrogen evolution experiments. Authors describe several experiments to determine the applicability of the material as a photocatalyst in HER reaction when assisted with Pt as a co-catalyst. In this regard, I am missing a full characterization of the material after the photocatalytic tests and some further (photo)(electro)chemical experiments:

1) what is the fate of the “molecular” Pt co-catalyst? HR-TEM measurements after photocatalysis could show if Pt NPs were formed during photocatalysis.

Response: This is a good suggestion. The $\text{H}_2\text{PtCl}_6 \cdot 6\text{H}_2\text{O}$ was reduced to Pt nanoparticles in the photocatalytic process, and these Pt nanoparticles act as co-catalysts. HR-TEM images of films after photocatalytic experiments are shown in **Fig. 6d - 6f**, which confirmed the formation of Pt NPs during photocatalysis from the high-angle annular dark-field (HAADF) TEM images and element mapping.

Fig. 6 *d* the HR-TEM of the film after photocatalytic experiment. *e* the HAADF-STEM images and *f* the EDS mapping; green: C; blue: N; red: Pt.

2) After some long-term cycles, I would perform the photocatalytic test with a fresh TEOA solution (without co-catalyst): if Pt NPs were formed, the material should still be active.

Response: We first did long-term cycles (100 h); after this cycle, fresh TEOA solution (without co-catalyst) was added, the materials were still active and showed stable HER performance for another 50 h, which confirmed the formation of Pt NPs.

Fig. 6b The long-term photocatalytic HER performance (100 h experiment; add fresh TEOA at 100 h for another 50 h photocatalytic experiment).

3) HR-TEM images after catalysis would show if the film was degraded during photocatalysis.

Response: We have now done the HRTEM analysis of the film after photocatalytic experiments. The carbon and nitrogen distribution from elemental mapping and energy dispersive spectrometer (EDS) with TEM revealed that both carbon (red) and nitrogen (green) atoms exist homogeneously in the skeletons. HRTEM images (Supplementary Figure 21 - 25) after photocatalytic experiments demonstrate the retention of crystalline domains.

The FI-IR and Raman spectra of the film before and after photocatalytic experiments showed a similar structure (Supplementary **Figure 19**, **Figure 20**). Both HRTEM, FT-IR, Raman spectra and long-term stable photocatalytic performance confirmed that the films were stable in the photocatalytic experiments.

Supplementary Figure 19 FT-IR of CTF film before and after photocatalytic experiment.

Supplementary Figure 20 Raman spectra of CTF films before and after photocatalytic experiment.

Supplementary Figure 22. TEM images of CTF film after photocatalysis. Bright field images and FFT images. Zone axis [001].

Supplementary Figure 24. TEM images of CTF film after photocatalysis. Bright field images and FFT images. Zone axis [010].

4) Instead of cycles, I would perform a single long-term photocatalytic experiment (+100 hours) to check the stability of the material.

Response: This is a good suggestion. We first did long-term photocatalytic experiment (100 h) as shown in **figure 6b**, the performance was very stable for 60 h, decreases after 60 h, but still active under reaction conditions till 100 h. And after adding some fresh TEOA into the reactor, and purging the reactor with N₂, then another 50 h photocatalytic experiment was carried out. It was still active and stable for 50 more hours; thus the film is active under reaction conditions as long as 150 h.

Fig. 6b. The long-term photocatalytic HER performance (100 h experiment; then fresh TEOA was added after 100h for another 50 h photocatalytic experiment).

Moreover, it is not clear the experimental procedure used in each cycle: new TEOA and Pt catalyst in each cycle?

Response: The detailed experimental procedure of cycling experiments: (1) The films catalysts were taken out, washed by deionized water thoroughly and dried at 60 °C; (2) fresh TEOA solution (10 mL TEOA + 90 mL deionized water) was added **without Pt catalyst** in the photocatalysis reactor; (3) the washed film catalysts were placed in the reactor; (4) The normal photocatalytic experiment was performed.

5) What happens when the reaction is performed without the addition of the Pt co-catalyst in the first cycle (or in a single 1-hour experiment)?

Response: We did a single 5-hour experiment without the addition of the Pt co-catalyst and the catalytic performance is shown in Fig. 6a. The performance without Pt is significantly lower than that of the film catalysts with Pt, and the HER rate of the film catalysts with Pt is about 32 times higher than that of the catalysts without Pt in the first 5 h.

Figure 6a. the HER performance of the film with and without Pt co-catalysts.

6) Is it possible to check the activity of the “film + Pt” material in an ITO electrode without light or TEOA and applying a potential to check the electrochemical properties of the material? I expect it to be very active: overpotential for HER could be obtained. Moreover, using a Clark electrode, faradaic efficiencies could be obtained, and this would show that the hydrogen evolution reaction is the only reaction taking place.

Response: We checked the electrochemical properties of the material by loading the “film + Pt” material on an ITO electrode without light or TEOA and applying overpotential vs. RHE from -0.5 V to 0.0 V. For the detection of H₂, GC was used instead of Clark electrode.

The overpotential was about 120 mV vs RHE (**Fig. 6c**), and the Faradaic efficiency was about 92 % when using a constant voltage (-0.35V vs. RHE), which confirmed the good selectivity for H₂ evolution (**Supplementary Table 7**). However, the current density was low because of the low Pt loading (2.0 wt % by ICP).

Fig. 6c Linear sweep voltammetry (LSV) of the CTF film with overpotential from -0.5

V to 0 V vs. RHE.

7) After photocatalytic experiments, and after thoroughly washing the film, ICP analyses of the film will show if undetectable Pt NPs (ultrasmall) or coordination compounds are retained in the film.

Response: As we explained in response to the first comment, the HRTEM images show the formation of Pt NPs on the film, that remained in the film even after thorough washing. We also did the ICP analyses of the film after photocatalytic experiments that show the presence of Pt in the film, and the amount of Pt was about 2.0 wt % based on ICP analysis.

8) A more detailed comparison with other similar systems should be added. Please, cite and compare the manuscript results with the results described in:

doi.org/10.3390/catal11060754

Response: This is a good suggestion; we have now provided a detailed comparison (supplementary **Table 4**) with the reported typical COFs or CTFs photocatalysts (**powder type**) as described in ref. (doi.org/10.3390/catal11060754). We have also provided a detailed comparison with similar system (**film-type**: g-C₃N₄ film; FS-COF film) (Supplementary **Table 5**). The relevant literature has also been cited the revised manuscript.

Minor general points:

1) Page 3: Sentence in Lines 32-34. Unfinished sentence? Or just an “and” missing?

Response: revised.

2) Page 4: Sentence in Lines 75-78. Even if I think that the meaning can be understood, it is difficult to read, and I would rephrase the sentence to make it more clear.

Response: revised.

3) Page 5. Line 102. “n-Hexane” should be written without capital letters.

Response: revised.

4) Page 6. Scheme 1. In the chemical reaction depicted in the scheme, I would add the by-products of the reaction in every step (water?).

Response: revised.

5) Page 6. Scheme 1. Line 107. “a) possible reaction mechanism”. I would rather describe them as “reaction steps” (or something similar) rather than “reaction mechanism”.

Response: revised.

6) Page 8. Line 132. Space missing at “be14.55”.

Response: revised.

7) Page 9. Sentence in Lines 155-157. Again, the sentence seems to be unfinished (or an “and” missing”.

Response: revised.

8) Page 9. Line 159. XPS description seems unfinished. It seems as if an initial paragraph is missing.

Response: Thanks much for pointing it out. We have now revised the description. The initial paragraph (experimental data: black curve) was not clear in the XPS data and we have now re-drawn the XPS curve. And the initial curve (experimental data) is shown using the **black circle (figure 3d, e)**.

Fig. 3 The X-ray photoelectron spectroscopy of the CTF film.

9) Page 9. Lines 163-164. Re-write the text in these lines.

Response: Thanks for your suggestion; the sentence has been revised.

10) Page 12. Line 217. “with Pt as cocatalyst”. Even if usually referred just as the cocatalysts, the role of Pt in these photocatalytic reactions is crucial and, at least, the actual Pt source (H_2PtCl_6) should be detailed here.

Response: We agree, it was not clear before and now we have revised it as below:

The photocatalytic hydrogen evolution activity of CTF film was investigated under visible light ($> 420 \text{ nm}$) with Pt nanoparticles (Pt NPs) as cocatalyst (Pt source is $\text{H}_2\text{PtCl}_6 \cdot 6\text{H}_2\text{O}$), and triethanolamine (TEOA) as a sacrificial agent.

All in all, I think that this manuscript should be accepted for publication if these comments are addressed (or refuted).

Response: We thank the reviewer for understanding and appreciating the significance of this manuscript again. We also appreciate the reviewers’ comments and suggestions that helped us to significantly improve the manuscript.

Reviewer #3:

In this paper, Tan et al. developed a smart synthetic strategy to first create a liquid/air interface, and fabricate successfully free-standing, large area, crystalline Covalent triazine frameworks (CTFs) films by this kind of interface. Furthermore, they explored

the application of the film as immobilized photocatalysts and showed that the fabricated CTFs film had an excellent performance of photocatalytic hydrogen evolution. From a practical point of view, immobilized CTFs are more suitable for large-scale water splitting applications, however, existing synthetic approaches mainly result in insoluble and non-processable powders, which makes their future device application still a huge challenge. It is indeed highly interesting in this work that an interface is created without a water phase, which provides a rational approach to prepare immobilized photocatalysts, which is a significant landmark for the development of CTF and COF films and their potential integration in future devices for applications in fields as photoelectric, sensing, separation, and energy technologies. After careful evaluation, I would like to recommend this paper for publication in Nature Communications after the authors address the following important points:

Response: We thank the reviewer for understanding and appreciating the significance of this manuscript.

1. Page 6, line 104: “Because of the weak polarity of the long carbon chain, the imine precursor was floating at the DMSO layer instead of dissolving in it”. Because this process is very important for the creation of liquid/air interface and the preparation of CTFs films, can the authors give further experiment details to describe this process?

Response: This is a good suggestion. We have provided the images of the imine precursor and aldehyde in DMSO (Supplementary **Figure 1**) to explain the process of the creation of the interface. As shown in Supplementary Figure 1, a clear interface was created between imine precursor and DMSO. However, the aldehyde was dissolved in DMSO to get a clear solution.

And the structure of the imine precursor was confirmed by ¹H-NMR (Supplementary **Figure 2**) and FT-IR (Supplementary **Figure 3**).

We have added the corresponding discussion as follows:

Because of the weak polarity of the long carbon chain, imine precursor float at the

surface of DMSO layer to generate an interface (Supplementary Fig. 1). The structure of precursor was confirmed by $^1\text{H-NMR}$ (Supplementary Fig. 2) and Fourier transform infrared (FT-IR) spectroscopy (Supplementary Fig. 3). Peak around 7.8 ppm can be assigned to C-H of benzene ring; peak around 8.3 ppm to C-H of imine bond and the peaks from 3.7 ppm to 0.9 ppm can be assigned to C-H of aliphatic chain (Supplementary Fig. 2). In FT-IR spectrum, peak of the amino group (3300 cm^{-1}) in hexylamine was disappeared, and the appearance of a new peak at 1645 cm^{-1} confirmed the formation of imine bond (Supplementary Fig. 3).

Supplementary Figure 1. Photographic image of the imine precursor and aldehyde monomer in DMSO. Imine precursor floating on DMSO (left), and aldehyde monomer dissolved in DMSO to get a clear solution (right).

Supplementary Figure 2. $^1\text{H-NMR}$ spectra of imine precursor in d-chloroform.

Supplementary Figure 3. FT-IR spectra of imine precursor, aldehyde, and hexylamine.

2. Interestingly, the CTF film is crystalline. As far as I know, the CTF powder prepared by the low-temperature poly-condensation method is amorphous (ref. 12). Can the author explain what reason causes this difference?

Response: The CTF powder firstly reported in (*Angew. Chem. Int. Ed.* 2017, **56**, 14149-14153) was amorphous. Recently, our group had developed two kinds of strategy to prepare crystalline CTF powder (*Adv. Mater.* 2019, **31**, 1807865; *Angew. Chem. Int. Ed.* 2018, **57**, 11968–11972; *Angew. Chem. Int. Ed.* 2020, **59**, 6007– 6014). We found decrease the monomer concentration is helpful to control nucleation and growth process in the preparation of crystalline CTF powder.

In this work, we were inspired by our own previously published works (*Adv. Mater.* 2019, **31**, 1807865; *Angew. Chem. Int. Ed.* 2018, **57**, 11968–11972; *Angew. Chem. Int. Ed.* 2020, **59**, 6007– 6014) and the imine-exchange strategy that published in *Science* (*Science*, 2018, **361**, 48-52). Here are the reasons for the formation of crystalline structure. (1) The imine precursor decreases the concentration of aldehyde monomer, which is helpful to control the nucleation (*Adv. Mater.* 2019, **31**, 1807865; *Angew. Chem. Int. Ed.* 2018, **57**, 11968–11972; *Angew. Chem. Int. Ed.* 2020, **59**, 6007– 6014); (2) The slow diffusion of monomer from the bulk solution to the interface may also decrease the concentration of monomers that participating in the reaction, and impact the growth process. (3) The reversibility of CTFs formation can be regulated utilizing the imine-exchange strategy (*Science*, 2018, 361, 48-52). So, we can obtain semi-crystalline film.

3. The XPS data (figure 2d) should be clarified further, the peak at 403 eV is not fitted in figure 2d, the authors should give more explanation.

Response: Thanks for your good suggestion. We have re-fitted the XPS data, and we added explanation in the revised manuscript.

Figure 3. The X-ray photoelectron spectroscopy of the film.

4. Did the authors do some special treatment of the films and glass support in the photocatalytic experiments, such as using some adhesive? The decrease of HER performance may be caused by the peel-off of film from the support.

Response: We did not do any special treatment of the films and glass support in the photocatalytic experiments. The films were directly loaded on the glass for HER experiments. We also proposed that the decrease of performance in the four cycles may be caused by the peeling-off of film, which would make weight loss and active area loss. And in a long photocatalytic experiment, the performance is stable for 60 h.

5. The experimental details of the SAXS and PXRD experiments should be described in the manuscript. Please include details of measurement conditions (i.e., powder or film), sample preparation, etc.

Response: We are sorry for missing the experimental details of the SAXS and PXRD experiments; now the details are added in the revised Supporting Information.

6. Some related literature about COFs films (10.1002/anie.202105190; Angew. Chem. Int. Ed. 2021, 60, 14875-14880; etc) can be added in the revised manuscript.

Response: Thanks for your suggestion. We have now cited these articles in the revised manuscript.

Reviewers' Comments:

Reviewer #1:

Remarks to the Author:

The manuscript from Hu et. al. has been revised satisfactorily, and authors have addressed the concerns from all the reviewers. The revised version of manuscript is ready to be published in Nature Communications, after considering following minor concerns:

1. There is a significant variation between the Lattice parameters calculated using DFT calculation, SAED and GIWAXS measurements (Supplementary Table 1). Authors should add an explanation.
2. Also, there is a big difference between theoretical and experimental elemental analyses (EDS). This could happen due to entrapped solvents or impurities present in the final CTF-materials.
3. While many COFs/CTFs show Apparent Quantum Efficiencies (AQE) about 2%, CTF films show very low values (maximum 0.11%). Authors should add an explanation.

Reviewer #2:

Remarks to the Author:

After checking the responses to the reviewers' comments and the modifications in the revised manuscript, I think that it should be accepted for publication in Nature Communications. The new experiments and results have answered all my previous doubts / questions.

Reviewer #3:

Remarks to the Author:

I have carefully gone through the author's response. Authors have answered all the questions satisfactorily and critically. I am recommending the manuscript for publication.

Given below are our responses (in BLUE colour) to the reviewer' comments.

Reviewer #1 (Remarks to the Author):

The manuscript from Hu et. al. has been revised satisfactorily, and authors have addressed the concerns from all the reviewers. The revised version of manuscript is ready to be published in Nature Communications, after considering following minor concerns:

1. There is a significant variation between the Lattice parameters calculated using DFT calculation, SAED and GIWAXS measurements (Supplementary Table 1). Authors should add an explanation.

Response: The variation between different measurements is a normal phenomenon (*Angew. Chem. Int. Ed.* **2020**, *59*, 6028-6036; *Sci. Adv.* **2020**, *6*, eabb5976), which may be caused by (1) systematic deviation: lattice parameters are affected by temperature or lattice distortion; (2) measurement deviation: variation in distance measurement of diffraction spots in SAED and the peak marking in GIWAXS. Typically, precise lattice parameters can be obtained via structural refinement based on the PXRD or single crystal XRD data.

Structure	a (Å)	b (Å)	c (Å)	Ref.
PI-2DP1 (DFT)	25.6	25.6	4.0	Angew. Chem. Int. Ed.
PI-2DP1 (SAED)	25.0	25.0	N.A.	Ed.
PI- 2DP1(GIWAXS)	27.0	27.0	3.9	2020 , 59 , 6028
L-2D-PI (DFTB)	25.9	25.9	N.A.	Sci. Adv. 2020 ,
L-2D-PI (SAED)	25.0	25.0	N.A.	6 , eabb5976
L-2D-PI (GIWAXS)	25.13	25.13	3.95	

2. Also, there is a big difference between theoretical and experimental elemental analyses (EDS). This could happen due to entrapped solvents or impurities present in the final CTF-materials.

Response: EDS is a qualitative or semi quantitative analysis method. Here, EDS was used to confirm if C and N elements were distributed uniformly in the sample. The

deviation of C and N analysis in EDS results may be caused by carbon-coated copper grids or the entrapped solvent (ethanol used in sample preparation of TEM).

3. While many COFs/CTFs show Apparent Quantum Efficiencies (AQE) about 2%, CTF films show very low values (maximum 0.11%). Authors should add an explanation.

Response: We added an explanation in revised *Supplementary Information*:

AQE was calculated based on the following equation that did not include the amount of catalyst, which may affect AQE value (Kazunari Domen. *Catalysis Letters*, **2015**, 145, 95-108). In our case, the film weight is about 1 mg, which is lesser than other COFs based systems (5 ~ 50 mg), so the AQE value in our case seems low. However, AQE (0.11%) is still comparable with N₃-COF (0.17; 400 nm; 10 mg; *Nat. Commun.* **2015**, 6, 8508)

$$AQE(\%) = \frac{2CN_A}{SPt\lambda/hc} \times 100\%$$

where C is the amount (μmol) of H₂ produced per hour; N_A is the Avogadro constant ($6.02 \times 10^{23}/\text{mol}$), h is the Planck constant (6.626×10^{-34} J/s), c is vacuum light velocity (3×10^8 m/s), λ is the monochromatic light wavelength (nm), t is the light irradiation time (1 h) and P is the incident monochromatic light intensity (W).